# Characteristic Features of Infrared Thermographic Imaging in Primary Raynaud’s Phenomenon

**DOI:** 10.3390/diagnostics11030558

**Published:** 2021-03-20

**Authors:** Lotte Lindberg, Bent Kristensen, Jane F. Thomsen, Ebbe Eldrup, Lars T. Jensen

**Affiliations:** 1Department of Nuclear Medicine, Copenhagen University Hospital, Herlev Hospital, Borgmester Ib Juuls Vej 1, 2730 Herlev, Denmark; bent.kristensen.01@regionh.dk (B.K.); lars.thorbjoern.jensen@regionh.dk (L.T.J.); 2Department of Occupational and Environmental Medicine, Copenhagen University Hospital, Bispebjerg and Frederiksberg Hospital, Bispebjerg Bakke 23, 2400 Copenhagen NV, Denmark; jane.froelund.thomsen@region.dk; 3Department of Endocrinology, Copenhagen University Hospital, Herlev Hospital, Borgmester Ib Juuls Vej 1, 2730 Herlev, Denmark; ebbe.eldrup@regionh.dk

**Keywords:** infrared thermographic imaging, primary Raynaud’s phenomenon, vibration white finger, diagnostic method

## Abstract

Raynaud’s phenomenon (RP) is characterized by the episodic whitening of the fingers upon exposure to cold. Verification of the condition is crucial in vibration-exposed patients. The current verification method is outdated, but thermographic imaging seems promising as a diagnostic replacement. By investigating patients diagnosed with RP, the study aimed at developing a simple thermographic procedure that could be applied to future patients where verification of the diagnosis is needed. Twenty-two patients with primary RP and 58 healthy controls were examined using thermographic imaging after local cooling of the hands for 1 min in water of 10°C. A logistic regression model was fitted with the temperature curve characteristics to convey a predicted probability of having RP. The characteristics time to end temperature and baseline temperature were the most appropriate predictors of RP among those examined (*p* = 0.004 and *p* = 0.04, respectively). The area under the curve was 0.91. The cut-off level 0.46 yielded a sensitivity and specificity of 82% and 86%, respectively. The positive and negative predictive values were 69% and 93%, respectively. This newly developed thermographic method was able to distinguish between patients with RP and healthy controls and was easy to operate. Thus, the method showed great promise as a method for verification of RP in future patients. Trial registration: ClinicalTrials.gov NCT03094910.

## 1. Introduction

Raynaud’s phenomenon (RP) is characterized by the episodic whitening of mainly the fingers and toes upon exposure to cold. Although the pathogenesis of RP is not completely understood, vascular, intravascular, and neural abnormalities are thought to contribute [1]. For a minority of patients, RP is secondary to certain drugs, exposure to hand-arm vibration, or an underlying rheumatic disease. However, for most patients, the condition is primary, in which case no underlying cause can be identified. Although troublesome, primary RP (pRP) is generally perceived as a benign condition that only affects the peripheral parts of the body and causes no permanent tissue damage [2]. Available treatments for pRP address symptoms and are not always effective. RP diagnosis is made based on the occurrence of symptoms after exclusion of secondary causes. However, a reliable diagnosis is crucial for patients with possible secondary RP (sRP) such as vibration white finger (VWF) after occupational exposure, in order for the condition to be acknowledged as an occupational injury and proper advice given concerning future job functions. Consequently, an objective measure is needed to verify the diagnosis made by the physician.

Presumably due to the complex etiology of RP, no gold standard method for verifying RP exists. Methods that are applied for verification comprise the finger systolic pressure (FSP) test invented in the 1970s [3]. The method measures the decrease in finger systolic blood pressure after total body and localized cooling of selected fingers. Sensitivity of the FSP test ranges from 51% to 92% and specificity from 81% to 100% [4,5,6,7,8,9,10], whereas some authors find it of no diagnostic use [11]. High sensitivity seems to occur with low cooling temperatures and increased disease severity. In our department, the FSP test is currently applied and has been used for decades. However, the method is cumbersome, time-consuming, and unpleasant for the patient. Moreover, it is performed with outdated equipment. Therefore, a new procedure for verifying RP needs to be developed.

Besides the FSP test, plethysmography, infrared thermographic imaging, and laser Doppler methods have been used to assess the RP diagnosis [2]. Plethysmography, which relies on finger volume change after venous occlusion, has not been widely accepted and reproducibility data are inconsistent. Laser Doppler methods assess the blood flow in the microcirculation by measuring the speed and concentration of red blood cells passing the area of interest. The methods have shown good reliability and correlation with infrared thermography [12]. However, the main role of the methods is still in research. Infrared thermographic imaging, which assesses the blood flow of the skin indirectly, has been thoroughly investigated and applied in patients with RP secondary to connective tissue disease (CTD), where it is mainly used to distinguish between pRP and RP secondary to systemic sclerosis (SSc). Its method applies temperature gradients as well as analysis of rewarming curves of the fingers obtained with infrared thermographic imaging after cold provocation [13]. The method was originally investigated to distinguish patients with RP from healthy individuals [14]. Previously, concerns about the reproducibility of thermographic imaging—especially after cold challenge—have existed, but a multicenter study concluded that the method had good reliability that was sufficiently high for use as an outcome measure in clinical trials [12]. Coughlin et al. applied infrared thermographic imaging to study the temperature gradients from fingertip to base in patients with hand-arm vibration syndrome [15] and were able to convincingly distinguish between the patient group and the control group without RP. As tempting as it may be to adopt and apply a previously described successful method, the particular method may not be suitable for routine diagnostic use or use in a different setting; the proportion of patients with RP in the referred population may vary considerably, especially between countries with different referral practices. 

The aim of the study was to investigate infrared thermographic imaging as a method to distinguish patients with RP from healthy controls. The thermographic procedure should be easy to perform, agreeable to the patient, and readily applicable in a clinical setting. The hypothesis was that if the method could be used to distinguish between patients already diagnosed with RP and healthy individuals, it could be applied to future patients where verification of the diagnosis is needed.

## 2. Materials and Methods

Twenty-two patients with pRP were recruited through advertisements in local newspapers. The diagnosis was clinically verified in accordance with the International Consensus Criteria for Primary Raynaud’s Phenomenon [16]. Participants were diagnosed with RP through the application of the three-step outline. Furthermore, subjects were included only if they met the following diagnostic criteria for pRP: (1) physical examination without findings suggestive of secondary causes, (2) no history of an existing CTD, and (3) negative antinuclear antibodies. Furthermore, nailfold capillaries of all 10 fingers were inspected (using a USB digital microscope, 25×–600× magnification) to document any abnormalities in nailfold capillary morphology that would indicate an underlying CTD. Participants were excluded if they reported any secondary causes of RP, such as treatment with a beta blocker or previous chemotherapy, exposure to vibration, or symptoms of carpal tunnel syndrome. 

The control group was recruited through advertisements in Herlev and Gentofte Hospital and in local newspapers. All 58 recruited subjects were healthy individuals who had no symptoms of RP or previous occupational exposure to vibration. They had no medical record of chronic illnesses or health issues affecting the blood vessels, especially heart or lung diseases, or CTDs. They did not have vitamin B_12_ deficiency or alcohol abuse, they were alleged non-smokers (one participant changed smoking status after inclusion), and they did not take prescriptive drugs. Thus, a total of 80 participants were examined using the thermographic method. In addition, the following blood analyses were performed on the included participants: hemoglobin, sodium, potassium, calcium (ionized), magnesium, thyroid-stimulating hormone (TSH), vitamin D, vitamin B_12_, folic acid, and HbA1c. TSH, electrolytes, and vitamins were analyzed to exclude competing causes of neuropathy, and HbA1c was analyzed to exclude diabetes. 

The examinations were performed in the Department of Nuclear Medicine, Copenhagen University Hospital, Herlev Hospital, from February 2017 to March 2018. 

### 2.1. Thermographic Imaging Procedure

On the day of examination, participants were asked to refrain from smoking, drinking coffee and tea, and performing strenuous exercise. Blood pressure was measured on both arms to exclude any side-to-side differences. 

Room temperature was 23 ± 1 °C. For acclimatization, the subjects stayed in the room for at least 30 min before examination. A thermographic photo was taken before the cold provocation to measure pre-cooling finger temperatures. Both hands were covered with thin plastic bags and immersed in 10 °C water for 60 s. Then, the plastic bags were quickly removed, and the hands were left to passively rewarm (palms down) on a towel. Video recording of the rewarming period was captured with an infrared thermographic camera (FLIR SC600, FLIR® Systems AB, Täby, Sweden). The video recording was discontinued when fingers approached pre-cooling temperature or after 60 min. Figure 1 shows the thermographic examination setup.

### 2.2. Data Preparation and Curve Analysis

Applying the software FLIR Research IR, version 3.3.12277.1002 (FLIR® Systems AB, Täby, Sweden), elliptical areas were drawn over the dorsal side of the middle phalanx of the 8 ulnar fingers, and a separate temperature curve was obtained for each finger. After the curves were cleared of artefacts, temperature curves were analyzed individually by using the R package *growthrates* to generate seven characteristics for each curve: lower lag time, slope of the curve during rapid rewarming (slope_rew_), upper lag time, temperature 50% through rewarming (t_50_), time to t_50_, end temperature (t_end_), and time to t_end_. Four additional temperature characteristics were noted or calculated: baseline finger temperature (t_base_), finger temperature immediately after cooling (t_0_), duration of thermographic recording, and the percentage of temperature recovery at end temperature (R% = temperature increase/initial temperature decrease) × 100%)) [14]. Figure 2 illustrates the curve and temperature parameters. 

### 2.3. Statistics

For descriptive statistics in Table 1, mean and standard deviation or median and interquartile range (IQR) were used where appropriate. Counts were shown as numbers and proportions. Group comparisons for normally distributed variables were conducted using Student’s t-test and with the Wilcoxon–Mann–Whitney test for non-normal variables. Group comparisons for categorical variables were performed using Fisher’s exact test due to the small cell counts. Due to the clustered nature of the data, a rank-sum test for clustered data was applied for comparing the hands [17] and showed no significant difference between the left and right sides. Thus, the data were reduced to single measurements per subject for modeling purposes. Multivariable logistic regression analysis was performed to assess which combination of variables could predict the presence or absence of RP. Figure 3 illustrates the thermographic procedure and subsequent analysis of the recording. The principles outlined by Steyerberg and Harrell [18,19] were applied when developing the clinical prediction model. Transformations or use of cubic splines were not deemed necessary for the continuous predictors. Predictor selection was based on clinical and purposeful knowledge, and if two predictors were highly correlated (Pearson’s *r* > ±0.75), only one predictor was selected. Backward elimination was used as a development tool, and model diagnostics and outlier detection were conducted according to suggestions by Fox and Weisberg [20]. Calibration was evaluated using a calibration belt and test for the calibration curve [21]. Internal model validity was assessed using bootstrap resampling with 300 samples. Performance of the final model was assessed using the calibration intercept and slope, and Brier score. The discrimination ability was evaluated using the concordance statistic (area under the curve (AUC) for the receiver operating characteristic (ROC) curve), along with threshold values for sensitivity, specificity, positive and negative predictive values, and accuracy. Further, the model was presented as a nomogram, generating a predicted probability of pRP. A cut-off level was derived from the most optimal point on the ROC curve. All statistical calculations were conducted using the statistical software R, version 4.0.2 (June 2020) and the following packages: *car*, *givitiR*, *growthrates*, *Hmisc*, *pROC*, *PropCIs*, *rms*, and *tableone*. Trial registration: ClinicalTrials.gov NCT03094910.

The study was approved by the Research Ethics Committee of the Capital Region of Denmark, protocol no. H-16035842, approved on 5 October 2016. Written informed consent was obtained from each participant prior to inclusion in the study. All procedures involving human participants were in accordance with the ethical standards of the institutional and/or national research committee and with the 1964 Helsinki Declaration and its later amendments or comparable ethical standards. 

## 3. Results

### 3.1. Clinical Characteristics

One healthy individual was excluded from the analyses, as the examination ended prematurely due to technical problems. Accordingly, the results from 79 subjects were included in the analyses. The study population consisted of 43 (54.4%) female and 36 (45.6%) male participants. Their clinical characteristics are presented in Table 1. None of the patients were treated for RP with drugs at the time of examination. Median duration of symptoms was 26 years (range: 1–64). Median (IQR) weekly episodes during winter months totaled 6 episodes (2;7) (range: 1–35); most of the patients did not have symptoms during the summer months. Except for significantly lower plasma potassium in the patient group compared with the control group (*p* < 0.04), blood test results showed no significant differences between the groups.

### 3.2. The Prediction Model

As the analysis of horizontally running curves did not generate interpretable results for *lower lag time*, *slope_rew_*, *upper lag time*, *t*_50_, and *time to t*_50_, a substitute *t*_50_* was calculated for each curve ((*t*_50_* = ((*t_end_* – *t*_0_) / 2) + *t*_0_). In addition, a binary variable describing the shape of the curve was added to replace the *slope_rew_* variable; each curve was named either S-shaped or horizontal. As a result, the following curve and temperature variables were applied in the subsequent logistic regression analysis: *t_base_*, *t*_0_, *curve type*, *t*_50_*, *t_end_*, *time to t_end_*, and *R%*. Age was also included in the analysis. Gender was not included due to the small number of male patients. The binary response variable of the logistic regression model was the diagnosis of RP made at the time of inclusion. Thus, all patients were positive for RP, and all healthy individuals were negative for RP. According to the fitted logistic regression model, the combination of predictors best able to predict the presence or absence of RP was *time to t_end_* and *t_base_*. The model specifics are presented in Table 2. Figure 4 presents the thermographic recordings from a healthy participant as well as a typical patient with RP and associated models of rewarming curves. The difference in *time to t_end_* is illustrated by the fact that the rewarming curve for the patient remains horizontal for a considerably longer time than for the healthy participant whose finger has returned to near-baseline temperature in less than 10 min. 

Diagnostics revealed no significant deviation from the model assumptions. Five outliers (both patients and controls) were reviewed, and as no irregular values were found, they were included in the analyses. The calibration plot and the calibration test showed that the intercept and the slope did not differ significantly from the ideal line (intercept = 0, slope = 1), *p* = 0.70, indicating that the predicted probability equals the actual (observed) probability. The calibration plot is presented in Figure 5. The AUC was 0.91 and the Brier score was 0.13 (Table 3). The Brier score assesses the accuracy of logistic regression models. It takes a value between zero and one. The lower the Brier score, the better the predictions are calibrated. The AUC value represents the ability of the model to make the correct diagnosis; the closer it is to one, the better the diagnostic ability.

### 3.3. Cut-Off Level 

Although the prediction model conveyed a predicted probability of having RP, the test result will ultimately depend on the chosen cut-off level, above or below which the diagnosis of RP is either confirmed or rejected, respectively. Based on the ROC curve generated from the model (Figure 6), the most optimal point on the curve (where both sensitivity and specificity were highest) yielded a sensitivity of 82% and specificity of 86%. The associated predicted probability cut-off level was 0.46. By use of this cut-off value, diagnosis was confirmed in 18 patients, while diagnosis was refuted in 4 patients. Diagnosis was correctly rejected in 49 healthy participants, whereas 8 healthy participants were falsely diagnosed with RP. The positive and negative predictive values (PPV and NPV) were 69% and 93%, respectively. Accuracy was 85%. A nomogram was generated so that prediction model results could be easily transferred into a useful outcome applicable in a clinical setting. The nomogram consisted of the predictor variables from the final logistic prediction model and reported a predicted probability of having RP (Figure 7). Figure 8 presents the results from the analysis of the rewarming curves, through the fitting of the prediction model to the chosen cut-off level and validation of the model.

## 4. Discussion

With thermographic video recording following cold provocation, temperature and curve characteristics were obtained and a prediction model was fitted with logistic regression. The calibration plot and the validation scores confirmed that the fitted prediction model was able to predict the presence or absence of RP. A nomogram was generated from the prediction model, reporting a predicted probability of having RP, and a cut-off level was found. In addition to being acceptable to the patient, the method generated clinically useful test results and proved to be applicable as a routine examination in a clinical setting. 

The currently applied FSP test is a well-established method that has been used for decades. The method performance is far from perfect and has been shown to vary according to factors such as symptom severity and extent of cooling [3,4,5,6,7,8,9,10,11]. The method has the drawbacks of being cumbersome, time-demanding, and unpleasant due to the cooling procedure. Thus, the aim was to find a diagnostic replacement, which should be less technically demanding, more pleasant to the patient, and not inferior to the FSP test in performance. Importantly, the method should also be readily applicable in a diagnostic setting. Investigation of thermography was chosen due to its non-invasive nature, its feasibility, and previous reports from the literature on the potential as a diagnostic procedure. Other methods such as laser Doppler methods also have the benefit of being non-invasive, but they have been reported to be less feasible than thermography [12] and have typically been investigated as a method to distinguish between pRP and RP secondary to SSc [22] and not as a diagnostic tool [23]. The sensitivity and specificity of the thermographic method developed in the present study were 82% and 86%, respectively, which are not inferior to the performance of the FSP test reported in the literature (sensitivity 51–92% and specificity 81–100%) [1,2,3,4,5,6,7,8]. Furthermore, the cold challenge procedure prior to the thermographic recording (local) was less extensive than that of the FSP test (total body). The thermographic procedure seemed less technically demanding than the FSP test, as the latter requires a tight operation of the technical equipment. Overall, the performance of the novel thermographic method was not inferior to the FSP test. In addition, the method seemed more feasible and more agreeable to the patient than the FSP test. 

Analysis of the thermographic images obtained with the present method was largely inspired by the work by O’Reilly et al. [14]. Their study was based on patients with pRP and sRP associated with SSc. Although they found that all the applied parameters differed significantly between patients and controls, the parameters were compared on a group basis. Similarly, *t_base_*, *t*_0_, and *R%* were significantly lower in the patient group than in the control group in our study (data not shown). Moreover, the patients showed a significantly longer *time to t_end_* than the healthy individuals. However, the variables showed considerable overlap between the groups. Only in combination were the variables shown to be able to predict the presence or absence of pRP on an individual basis.

Schuhfried et al. and Cherkas et al. [24,25] also applied thermographic temperature parameters to fit a prediction model with logistic regression. Both applied data derived from patients with RP secondary to CTDs and found that only pre-cooling parameters were able to predict the presence of RP. However, the discriminatory power was relatively low, with a sensitivity of 11% and specificity of 96% [25]. The present study also found the baseline temperature to be a predictor of pRP. However, our final model included the dynamic parameter time to t_end_, which was not investigated in the mentioned studies. Contrarily, none of the dynamic parameters were found to be significant predictors by Schuhfried et al. [24]. Their prediction model found patients with definite RP (sensitivity and specificity of 77% and 73%, respectively) but was not able to identify the patients with only probable RP (sensitivity and specificity of 5% and 95%, respectively) [24]. Their reported “definite RP” test characteristics are similar to the characteristics found in the present study. Lim et al. [26] investigated baseline temperature differences between the palm of the hand and the coolest finger and found them to be predictive of pRP, although with test results inferior to the results found in the present study (sensitivity 67%, specificity 60%). 

As in our study, von Bierbrauer et al. and House et al. [27,28] found the time of rewarming to be diagnostic of RP secondary to vibration exposure. Moreover, neither baseline nor cooling absolute temperatures were found to be useful diagnostic tools [27]. A sensitivity of 67%, a specificity of 61%, and an AUC of 72% were reported [28]. The differences in performance compared to our method could be due to differences in the cooling method or the rewarming time period, or the fact that the mean temperature difference was the only parameter reported besides the rewarming time. Their method did not include as many parameters as our newly developed method.

Coughlin et al. also investigated infrared thermography in patients with RP secondary to vibration exposure and controls [15]. Their thermographic method proved to be extraordinarily successful at discriminating between patients and controls, yielding a sensitivity of 100%, a specificity of 88%, a PPV of 95%, and an NPV of 100% after cold provocation. Accordingly, this method performed better than the present. However, the examinations by Coughlin et al. were performed at a tertiary referral center with, presumably, a high proportion of patients with severe RP. In addition, the rewarming period took place in an environmental chamber with regulated air temperature, air velocity, and humidity. Such conditions may very well be required [2,29] to enable such a distinct discrimination between patients and controls using infrared thermographic imaging. However, an environmental chamber is not routinely available in diagnostic settings. In contrast to the method by Coughlin et al., our thermographic method aimed at being applicable in a routine clinical setting. Furthermore, their findings could be difficult to reproduce in a setting with lower prevalence of RP and less severe symptoms.

Studies on thermographic imaging in patients with RP tend to appear heterogeneous in terms of RP origin, extent of cold challenge, and method of image analysis. However, it seems that baseline temperatures are generally better as a discriminator in patients with pRP or RP associated with rheumatic disease than in patients with VWF. Rheumatic diseases such as SSc cause structural changes to the tissue, which may affect the basal blood flow to the fingers [1] and thereby lower their baseline temperature. However, structural tissue change is absent in pRP by definition [2]. Even so, patients with pRP may have either an intravascular or a neural defect that permanently affects the temperature of their hands, suggesting that pRP is, at least partly, a disease with chronic changes and not entirely a dynamic disease. Accordingly, both t_base_ (static parameter) and time to t_end_ (dynamic parameter) were predictors of pRP in the final model, although the dynamic parameter contributed more to the model than t_base_. 

The various methods of image acquisition, image processing, and subsequent data processing make it challenging to compare studies on thermal imaging. Typically, image processing includes selection of areas of interest and possibly temperature gradients. The use of temperature gradients from acral to more proximal parts of the hand depends on the type of RP in question, as the distal–dorsal difference has been shown to be able to distinguish pRP from RP secondary to SSc [13,30]. However, gradients are not necessarily a good predictor of the condition where structural tissue changes are not a prominent part of the condition such as in VWF [31]. Laser speckle contrast imaging (LSCI) and laser speckle contrast analysis (LASCA), which are types of laser Doppler methods, measure the blood perfusion of the skin. The image processing resembles that of thermographic imaging [12,32]. The advantage of the image processing for both laser Doppler methods and thermographic imaging is that it is easily comprehensible and simple to perform. Although a study suggested systematic differences between observers [12], another study reported a good inter-rater reliability in thermographic imaging [30]. Accordingly, the disadvantages of this type of image processing are the inter-observer difference, the time consumption, and the need for image analysis training. In recent years, methods applying machine learning (ML) for thermal image processing in medical imaging have intensified [33]. The use of ML has been suggested to be able to assist the clinical evaluation of rheumatoid arthritis [34], the diabetic foot [35], and breast cancer [36]. However, the use of ML in the assessment of carpal tunnel syndrome—which can mimic RP—only proved effective in severe cases [37]. The use of thermography and ML in the detection of RP has been suggested recently [38] but has not yet been validated. The advantage of the computer-assisted image processing is the fast, objective, and potentially fully automated evaluation. However, the use of artificial intelligence to support the clinical decision making requires large databases [39], and the use of ML in RP diagnostics is currently limited by the lack of infrared thermal imaging databases [38]. 

The study included 22 patients with pRP—examined and diagnosed by one investigator applying the same international consensus criteria, making the diagnostic criteria reliable and consistent throughout the inclusion—and 58 healthy individuals without RP or related conditions. Only few data were missing. The lack of multiple measurements for each participant is a limitation to the study. Thus, the repeatability is not possible to assess from the current data. However, a multicenter study [12] reported that the reliability of thermography was sufficiently high for use as outcome measures in clinical trials, and other studies have shown good intra-individual reproducibility [26,40]. Surely, further investigation of the present thermographic method requires repeated measurements and evaluation of the test–retest reliability as well as inter-observer agreement. In addition, the possibility of misclassification exists, although the prediction model was based on the diagnosis of pRP by interview, inspection, and blood test results to exclude underlying causes. Before the novel method can be put into practice, the method should be tested in future patients who are referred for verification of RP. Furthermore, investigation of the test–retest reliability of the method would be advantageous.

## 5. Conclusions

A novel thermographic method for verifying RP was developed based on patients with pRP and healthy controls. The thermographic images before and after cold provocation were analyzed, and the resulting temperature characteristics were used to fit a logistic regression prediction model. Goodness of fit and internal validation were evaluated. A cut-off level was proposed based on the ROC curve. Furthermore, a nomogram was constructed from the final prediction model to make the method available for a clinical and diagnostic setting. The presented method based on thermographic imaging and subsequent newly developed data analysis proved to be able to distinguish between patients with RP and healthy individuals. Importantly, the method was found to be agreeable to the patient and highly applicable in a clinical setting. As a result, the presented method seems useful as a method to verify RP such as VWF in future patients, where verification of RP is needed. The method should be tested in future patients of the target population before it can be applied as a routine examination.

## Figures and Tables

**Figure 1 diagnostics-11-00558-f001:**
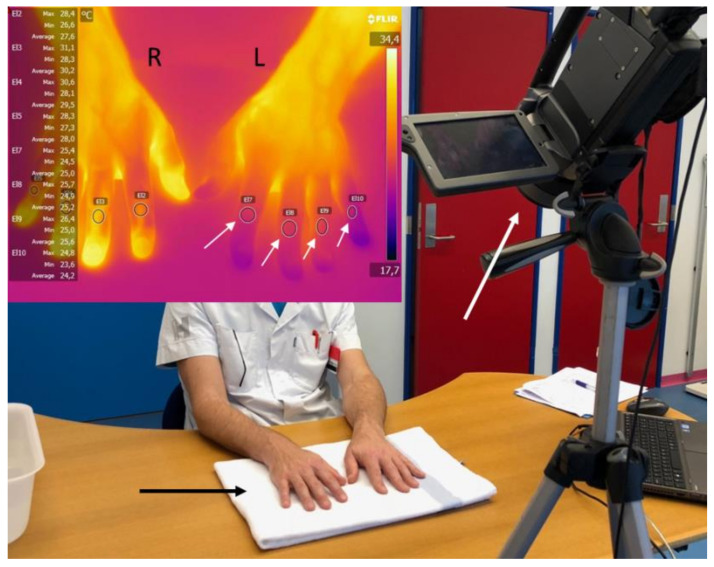
The thermographic examination setup with the infrared thermographic camera (large white arrow). R = right, L = left. After the cold challenge, the hands were left to rewarm on a towel (black arrow). The thermographic image illustrates the hands during rewarming. The small white arrows mark the elliptical area of interest on the middle phalanx of each investigated finger on the left hand. Both hands were investigated.

**Figure 2 diagnostics-11-00558-f002:**
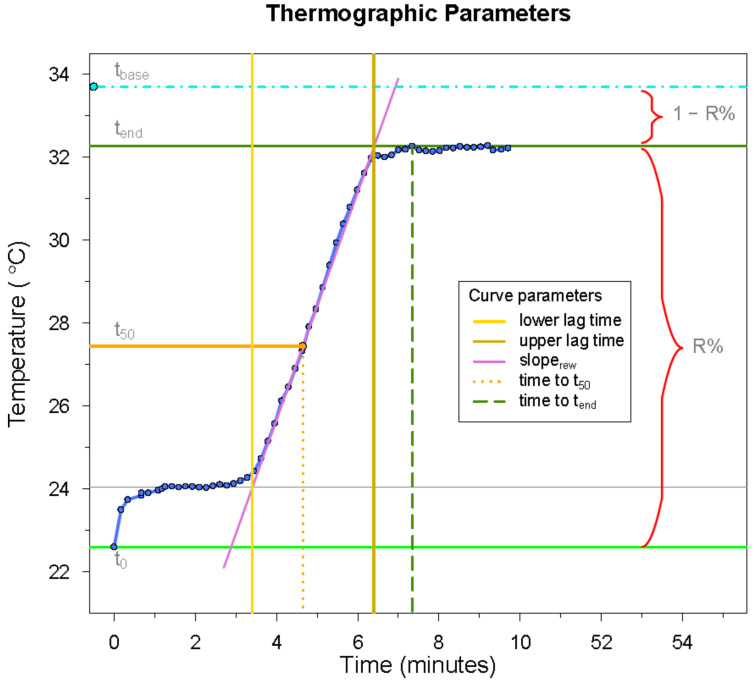
The thermographic parameters initially included in the curve analysis. The S-shaped curve is shown as an example, as the initial analysis on horizontally running curves was unsuccessful for some of the parameters. The light blue “dot-dashed” line marks the baseline finger temperature (*t_base_*), the dark green line marks the end temperature (*t_end_*), and the percentage temperature recovery from the first to the latter is the *R%*. The vertical “long-dashed” line represents the *time to t_end_*. The orange line marks the finger temperature halfway through rewarming (*t*_50_), while the vertical dotted line marks the *time to t*_50_. The vertical yellow and gold lines represent the lower and upper lag times, respectively. The green horizontal line marks the finger temperature immediately after cooling (*t*_0_).

**Figure 3 diagnostics-11-00558-f003:**
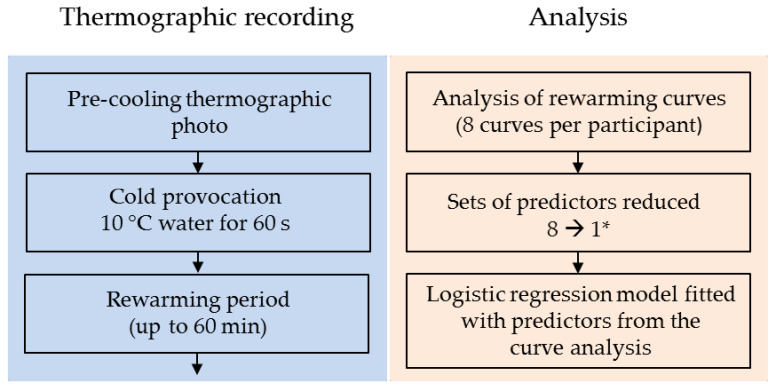
Flowchart of the proposed acquisition of rewarming curves after cold provocation (blue section) and the subsequent analysis of the curves, including the fitting of the prediction model (orange section). *The analysis conveyed eight sets of predictors, one set for each of the four ulnar fingers on each hand, which were reduced to one set by selection of the following values: *t_base_* (mean), *t*_0_ (mean), *t_end_* (lowest value), *time to t_end_* (highest value). *t*_50_* and *R%* were calculated from the selected values of *t_base_*, *t*_0_, and *t_end_*. *Curve type* was classified as S-shaped if all curves were S-shaped. If one or more curves were horizontal, *curve type* was classified as such. *t_base_* = baseline finger temperature; *t*_0_ = finger temperature immediately after cooling; *t_end_* = end temperature; *t*_50_* = finger temperature halfway through rewarming; *R%* = the percentage temperature recovery at end temperature.

**Figure 4 diagnostics-11-00558-f004:**
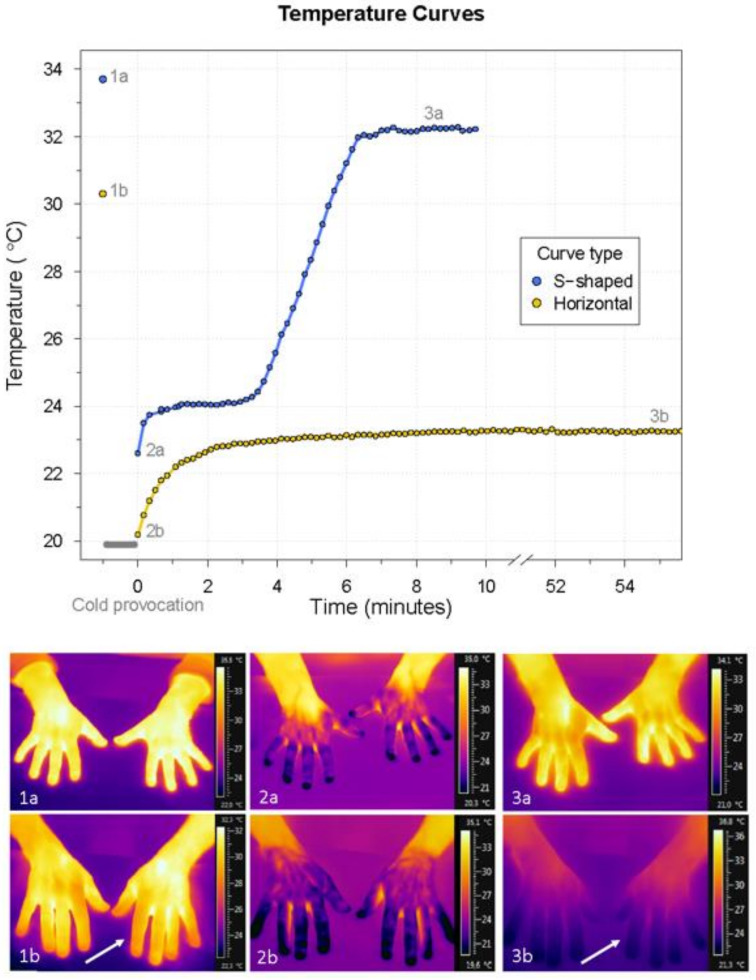
The two curve types identified during the analysis of the thermographic temperature curves. The numbers 1–3 refer to the time points of baseline, immediately after cooling, and at the end of the thermographic examination, respectively. These time points correspond to the temperature variables *t_base_*, *t*_0_, and *t_end_*, respectively. The thermographic images associated with the mentioned time points are shown. The gray bar marks the time of the cold provocation. The top bar of the thermographic images shows the hands of a healthy participant (**a**), while the bottom bar presents the hands of a patient with RP (**b**). Note the lower temperature in the fingers of the patient compared with the healthy participant, especially at baseline and at end temperature (arrows). *t_base_* = baseline finger temperature; *t*_0_ = finger temperature immediately after cooling; *t_end_* = end temperature.

**Figure 5 diagnostics-11-00558-f005:**
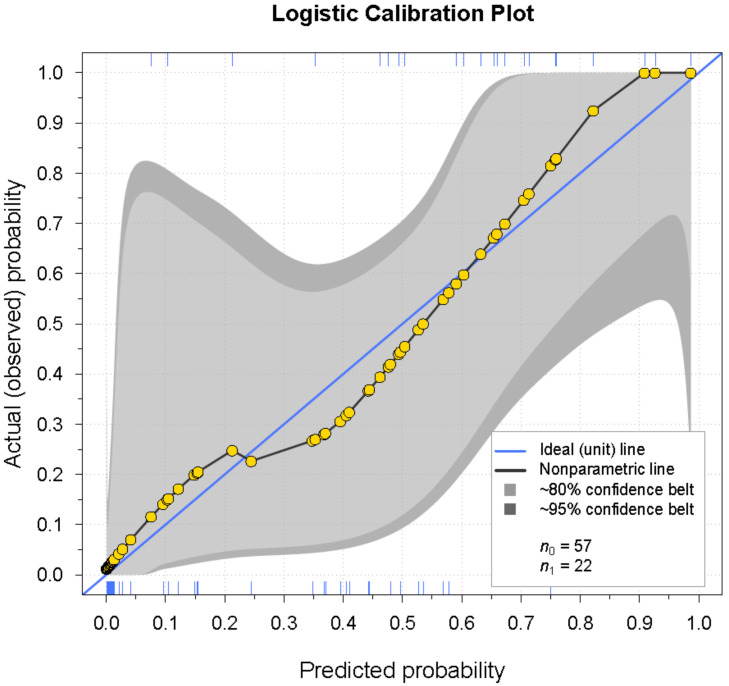
Calibration belt plot with 80% and 95% confidence intervals. The blue diagonal line represents the ideal calibration line, where predicted probability equals actual (observed) probability. The black line with the gold observation points represents the fitted logistic regression model. The model did not differ significantly from the ideal line, *p* = 0.70. The rugs at the bottom and top of the plot give an impression of the density of the predictions for controls and patients, respectively. The calibration belt was plotted according to the method described in [21].

**Figure 6 diagnostics-11-00558-f006:**
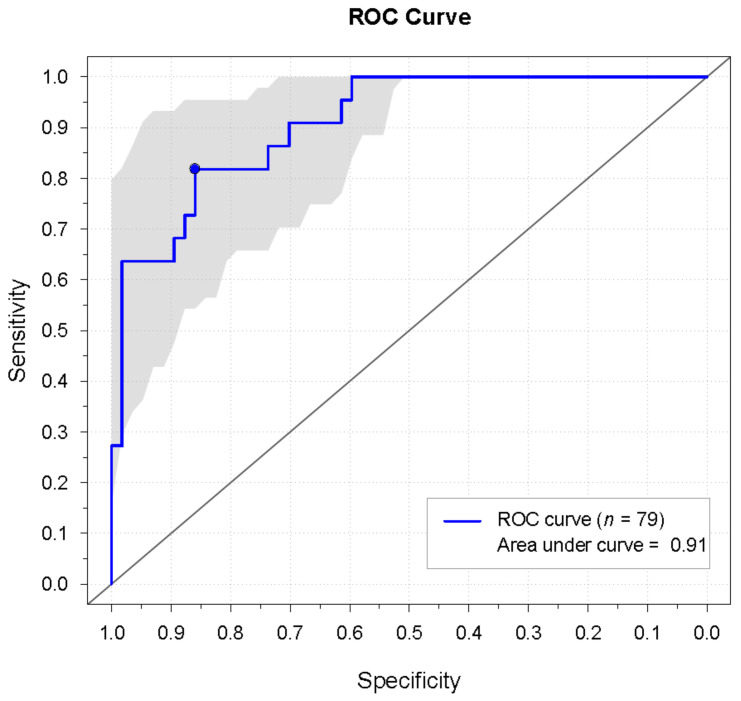
The receiver operating characteristic (ROC) curve generated from the final logistic regression model. The blue point marks the place on the curve where sensitivity and specificity are highest. This optimal point on the curve was used to derive the cut-off value, above or below which diagnosis is either confirmed or rejected, respectively. The gray area surrounding the ROC curve represents the 95% confidence interval.

**Figure 7 diagnostics-11-00558-f007:**
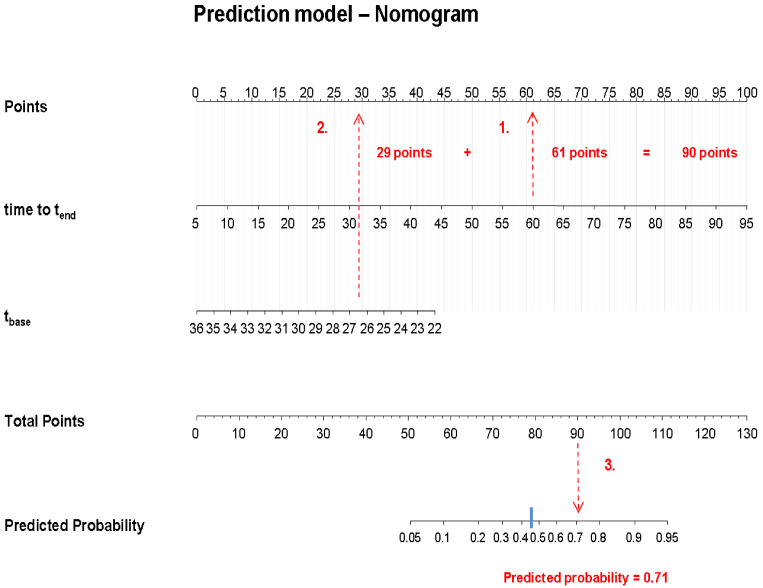
The nomogram constructed from the final logistic regression model generated from the described thermographic procedure. The red illustrations mark an example of how to read the nomogram (a set of values generated from a patient); *time to t_end_* = 60 min., *t_base_* = 26.5°C. In this example, the predicted probability of RP is above the cut-off value of 0.46 (blue line) and according to the thermographic test, the patient is positive for RP. For predicted probabilities below the cut-off value, diagnosis will be rejected. *t_base_* = baseline finger temperature, *time to t_end_* = time to end temperature.

**Figure 8 diagnostics-11-00558-f008:**
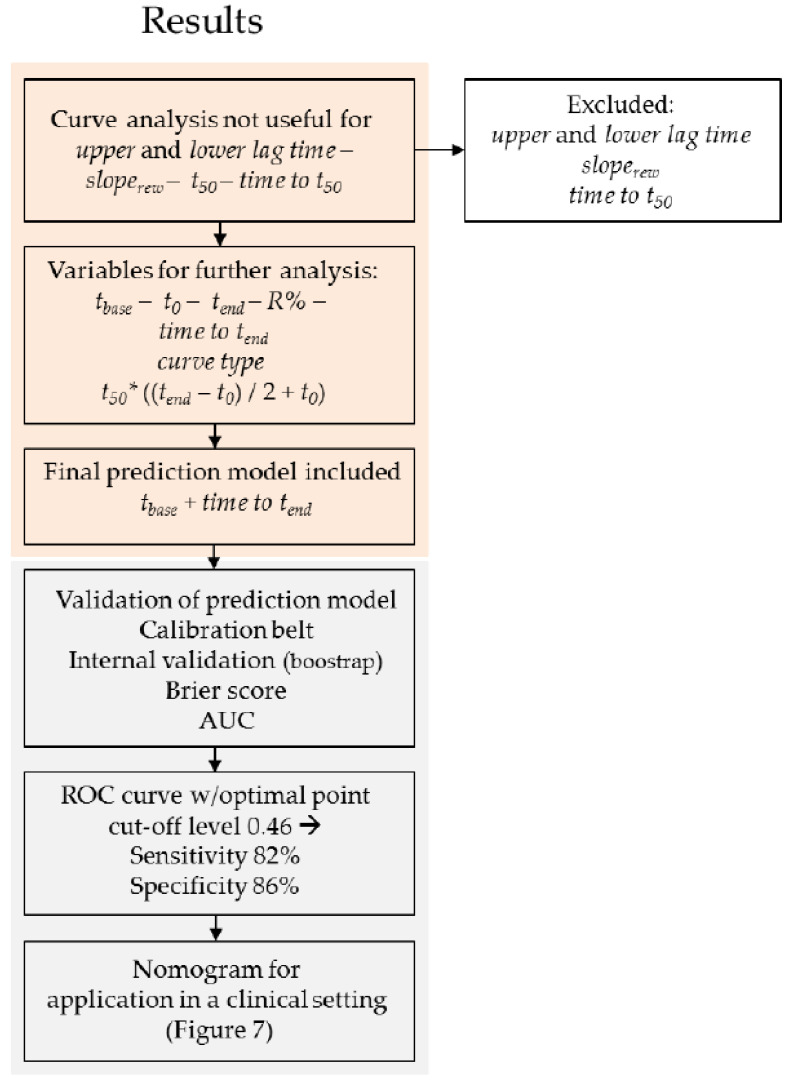
Flowchart of the results generated from the analysis of rewarming curves, fitting of prediction model (both orange section), and validation of the model as well as the subsequent calculation of the proposed cut-off level and construction of the nomogram (gray section). The orange section corresponds to the orange section of Figure 3. *slope_rew_* = slope of the curve during rapid rewarming; t_50_ = finger temperature halfway through rewarming; *t_base_* = baseline finger temperature; *t*_0_ = finger temperature immediately after cooling; *t_end_* = end temperature; *R%* = percentage temperature recovery at end temperature; *t*_50_* = finger temperature halfway through rewarming (calculated). AUC = area under the curve. ROC = receiver operating characteristic.

**Table 1 diagnostics-11-00558-t001:** Clinical characteristics of all participants. Data are presented as number (%), mean (SD), or median (IQR). pRP = primary Raynaud’s phenomenon.

		Patients, pRP	Controls	*p*-Value
Number of participants		22	57	
Gender	Female	19 (86.4)	24 (42.1)	<0.001
	Male	3 (13.6)	33 (57.9)	
Age		57.2 (10.0)	57.8 (12.5)	0.82
Smoking status	Never	11 (50.0)	35 (61.4)	0.51
	Current	0 (0.0)	1 (1.8)	
	Former	11 (50.0)	21 (36.8)	
Tobacco (pack-years)		0 (0;5)	0 (0;3)	0.81
Alcohol (units/week)		3 (1;7)	4 (2;7)	0.83

**Table 2 diagnostics-11-00558-t002:** The predictors included in the final logistic prediction model. Estimate is the coefficient of the predictor variables with their respective standard errors. *p*-Value describes the respective significance levels of the variables in the model.

Predictor	Estimate	Std. Error	Wald χ2	*p*-Value
Intercept	2.4	4.9	0.50	0.62
time to t_end_	0.11	0.04	2.9	0.004
t_base_	–0.30	0.15	–2.0	0.04

**Table 3 diagnostics-11-00558-t003:** Calibration and internal validation test results before and after bootstrapping (samples *n* = 300). The area under the curve (AUC) is presented with a 95% confidence interval.

Final Model	Original	Bootstrap Corrected
Calibration intercept	0.00	–0.01
Calibration slope	1.00	0.89
Brier score	0.11	0.13
Concordance statistic/AUC	0.91 (0.84–0.98)	-

## Data Availability

The data presented in this study are available on request from the corresponding author.

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
