# Peer review of "Characteristic Features of Infrared Thermographic Imaging in Primary Raynaud’s Phenomenon"

_diagnostics, 2021, doi:10.3390/diagnostics11030558_

Round 1
Reviewer 1 Report
-English should be corrected
-please add colorful picture of measurements (optionally);;; + arrows what is what
-please add block diagram of the proposed research step by step ;;; what is the result of paper?;;;
-please add block diagram of the proposed method;;;
-please add photo/photos of application of the proposed research ;;;;
-please add sentences about future analysis;;;
-Figures should have better quality;;;;
-Fonts of figures should be bigger;;;
-please add arrows to photos what is what;;;
-formulas and fonts should be formatted;;;;
-please add labels to figures;;;
-references should be 2018-2021 Web of Science about 50% or more ;; 30 at least
-Please compare with other methods, justify. Advantages or Disadvantages different methods
for example about image recognition/thermal imaging
1) Fault diagnosis of electric impact drills using thermal imaging, Measurement, Volume 171, 2021,
https://doi.org/10.1016/j.measurement.2020.108815
-Conclusion: point out what are you done;;;;
-is there possibility to use the proposed methods for other problems?
Reviewer 2 Report
This work is well within the scope of Diagnostics and it may be of interest to most of the readers of this journal. It is well organized, with few and rather old, references to follow.
I would expect to see some more discussion regarding the state-of-the-art practices used for RP patients, besides thermographic imaging.
Furthermore, as authors stated in the discussion section, ‘studies on thermographic imaging in patients with RP tend to appear heterogeneous in terms of RP origin, extent of cold challenge, and method of image analysis’ thus someone would expect to see a study including greater number of patients for safer conclusions. The current study was conducted from February 2017 to March 2018 and included 22 patients with pRP and 58 healthy individuals without RP or related conditions. Furthermore, it is stated that ‘Only few data were missing. The lack of multiple measurements for each participant is a limitation to the study’. It is not very clear why the study was not extended until 2020?
Specific comments
Abstract
‘The current verification method is outdated, but thermographic imaging seems promising as a diagnostic replacement.’ What other methods could be used? Please provide a comparison in terms of sensitivity and specificity. Can authors discuss on the state-of-the-art methods in the introduction section?
‘The characteristics time to end temperature and baseline temperature were the most appropriate predictors of RP’ ‘Among the examined’ could be added here for clarification.
‘AUC’ Please define every abbreviation the first time cited in the text.
Introduction
P2, L46,47: ‘In our department, finger systolic pressure with cold provocation is currently applied and has been used for decades [3]. The method is cumbersome, time-consuming, and unpleasant for the patient’ Please discuss on the state-of-the-art practices used for RP patients.
Discussion
P10, L349, 50: ‘Only few data were missing. The lack of multiple measurements for each participant is a limitation to the study. Thus, the repeatability is not possible to assess from the current data’. Since the repeatability is not possible how the authors support their conclusion ‘… the presented method seems very useful as a method to verify RP such as VWF in future patients, where verification of RP is needed’ ? (P10, L362)
Round 2
Reviewer 1 Report
The paper is better,
-Figure 1 - please add arrows what is what
-Figures should have better quality;;;;
-references should be 2016-2021 Web of Science about 50% or more ;; 30 at least.
-Impact Factor is computed for 2-5 last years (plese read about IF), papers < 2015 are old.
you should show new knowleddge 2016-2021
-Please compare with other methods, justify. Advantages or Disadvantages for image processing;;;
for example
1) Recognition of images of finger skin with application of histogram, image filtration and K-NN classifier, Biocybernetics and Biomedical Engineering, 2016,
https://doi.org/10.1016/J.BBE.2015.12.005
etc.
write keywords in Scopus/WoS and compare with new papers about similar topics.
Reviewer 2 Report
My previous remarks were addressed
Author Response
Please see the attachment regarding comments for Reviewer 1.